# Thought Cloning: Learning to Think while Acting by Imitating Human Thinking

**Shengran Hu**[1,2]
srhu@cs.ubc.ca

**Jeff Clune**[1,2,3]
jclune@gmail.com

[1]Department of Computer Science, University of British Columbia
[2]Vector Institute
[3]Canada CIFAR AI Chair

## Abstract

Language is often considered a key aspect of human thinking, providing us with exceptional abilities to generalize, explore, plan, replan, and adapt to new situations. However, Reinforcement Learning (RL) agents are far from human-level performance in any of these abilities. We hypothesize one reason for such cognitive deficiencies is that they lack the benefits of thinking in language and that we can improve AI agents by training them to *think like humans do*. We introduce a novel Imitation Learning framework, Thought Cloning, where the idea is to not just clone the behaviors of human demonstrators, *but also the thoughts humans have as they perform these behaviors*. While we expect Thought Cloning to truly shine at scale on internet-sized datasets of humans thinking out loud while acting (e.g. online videos with transcripts), here we conduct experiments in a domain where the thinking and action data are synthetically generated. Results reveal that Thought Cloning learns much faster than Behavioral Cloning and its performance advantage grows the further out of distribution test tasks are, highlighting its ability to better handle novel situations. Thought Cloning also provides important benefits for AI Safety and Interpretability, and makes it easier to debug and improve AI. Because we can observe the agent's thoughts, we can (1) more easily diagnose why things are going wrong, making it easier to fix the problem, (2) steer the agent by correcting its thinking, or (3) prevent it from doing unsafe things it plans to do. Overall, by training agents *how to think* as well as behave, Thought Cloning creates safer, more powerful agents.[1]

## 1  Introduction

Language may be the key to what separates humans from all other animals, endowing us with an amazing level of general intelligence [1–4]. Crucially, the benefits of language are not confined to improving our ability to communicate with others: language also helps us *think* better [2–4]. We first describe the benefits of agents that can *understand* language (a common topic in AI) before moving to the benefits of agents that *think* in language (a topic that has received far less attention).

There are many benefits that arise if our agents can understand language. Doing so is crucial for agents to generalize to new tasks we want them to perform. This is because it is drastically more sample efficient if one can tell an agent what the task is, rather than requiring the agent to figure out the task through trial and error [5, 6]. Moreover, agents that can understand language allow us to define new tasks at test time without having to anticipate every wish we might eventually have for

---

[1]The source code, model weights, and dataset are available at `https://github.com/ShengranHu/Thought-Cloning`.

37th Conference on Neural Information Processing Systems (NeurIPS 2023).

our trained agents [7]. That is in contrast to conventional hand-designed task descriptions, which can be vast, but still place constraints on what we can ask an agent to perform [8].

While the benefits of agents that can understand language are commonly discussed, there has been relatively little discussion in AI, especially in Reinforcement Learning (RL), regarding the many benefits of agents that *think in language*. While it remains debated whether humans exclusively think in language [9, 10], many scholars believe that natural language is intricately woven into our thought processes, conferring distinct cognitive advantages: thinking in language helps humans generalize, extrapolate, adapt to new situations, combine old knowledge in new ways, explore, plan, replan when necessary or beneficial, transfer knowledge via analogies, and the list goes on [2–4]. Despite these benefits, AI agents rarely, if ever, *think*, at least not in human language. While neural networks have internal vector activations that can be considered thinking, many hypothesize that there are specific benefits to thinking in the discrete, symbolic form of language (e.g. combining ideas in an exponential number of ways) [6, 11, 12], meaning that agents that think in language might learn faster, perform better, and generalize better than non-lingual agents.

In addition to agents being more capable, there are major benefits regarding AI Safety and Interpretability that arise when agents think in our language. If one can watch an agent think during training, one can recognize deficiencies in skills or values that can be improved, or one could decide the agent is not ready to be deployed. During testing, one can constantly scan the thoughts of the agent and intervene when the agent plans to do something undesirable. For example, if an agent thinks "My goal is to take my passenger to the store as fast as possible so I will run through this red light without stopping" one could intervene to stop that behavior ahead of time. Furthermore, watching agents think enhances the steerability of agents. If an agent is confused when solving challenging tasks, one can inject their thoughts into the agent to help it solve the task in a desired way. A final major benefit of agents that think in human language is it makes it easier to train more capable, safer AI agents. One can spot *why* things are not working, instead of just seeing that they are not working, and that provides ideas for how to debug and or improve AI training.

For all these reasons, adding the ability of AI agents to think in language could produce many significant advantages, and we suggest that the most effective way to achieve this goal is by *imitating human thinking*. Humans do not acquire thinking skills in isolation; instead, they are learned in part through demonstrations and feedback provided by teachers [2, 13–15]. As such, a promising method is to have agents learn from demonstrations where humans think out loud while acting. This approach is distinct from existing works that leverage pre-trained Large Language Models (LLMs) for planning [16, 17], because such LLMs are not trained on data where humans think out loud *while acting*. Thought data, such as YouTube videos and transcripts [18, 19], contains millions of hours of people talking out loud while performing tasks, revealing the thinking behind their actions, planning, decisions, and replanning, such as when they play video games [19]. This thought data is greatly valuable and widely available (Section 2), but has not yet been extensively explored, and this work hopes to encourage further research into the use of thought data to teach thinking skills to agents.

Provided we can solve the real, significant challenges of AI Safety and existential risk [20–24], there are tremendous gains to be had by creating more powerful AI or even AGI. In this paper, we propose a novel Imitation Learning framework, Thought Cloning, where agents not only learn to act from human demonstrations, as in Behavioral Cloning [25], but also *learn to think* from demonstrations where human think out loud while acting. Although we expect Thought Cloning to truly shine when trained on vast online datasets of synchronized human thoughts and actions, this paper validates the concept with synthetic thought data in a challenging domain, BabyAI [26]. Our experimental results illustrate that Thought Cloning outperforms Behavioral Cloning, even when Behavioral Cloning agents have the ability to think (in latent vectors), but have to learn that skill without the supervision of thinking provided by Thought Cloning. We also demonstrate that Thought Cloning generalizes better than Behavioral Cloning in out-of-distribution tasks in both zero-shot and fine-tuning settings. Finally, we provide empirical evidence for the previously discussed advantages of Thought Cloning in terms of Safety and Interpretability, where unsafe behavior can be near perfectly stopped before execution. All told, the results are promising and offer a glimpse of the enormous potential of Thought Cloning to not only make AI smarter, but also safer and more interpretable.

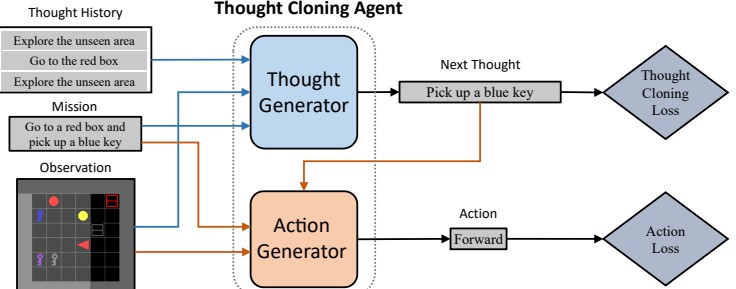

Figure 1: Overall framework for Thought Cloning (TC). The TC agent has two components: the Thought Generator and Action Generator. At each timestep, the TC agent receives an observation, a mission, and a history of thoughts as inputs. The Thought Generator generates thoughts, and the Action Generator generates actions conditioned on these thoughts. Generated thoughts and actions are compared to the ground truth from the demonstration dataset to calculate the loss.

## 2 Proposed Method

Conventional Imitation Learning methods [27, 28], such as Behavioral Cloning [25], strive to construct a policy that accurately replicates the distribution of behavior in a given dataset of demonstrations. However, our proposed framework, Thought Cloning, diverges from this approach by aiming to teach agents how to also *think* while acting, utilizing a synchronized dataset of human thinking. The thought dataset, denoted as $\mathcal{D} = \{D_i\}_{i=1}^N$, comprises a series of trajectories, $D_i = (m, \{(o_t, th_t, a_t)\}_{t=1}^T)$. Each trajectory encompasses a mission, $m$, defined in natural language, along with an observation $o_t$, an action $a_t$, and a corresponding thought $th_t$ at each timestep, $t$. Such datasets are widely available online. For example, by inferring action labels from Youtube videos with VPT [19] and then retrieving the corresponding transcripts, we can obtain a thought dataset that contains both human thinking and action [19, 18]. In such a dataset for Minecraft, a thought like "I need to gather wood to build a shelter before nightfall" might correspond to the player moving towards a tree and collecting wood. It is worth noting that noise is an inevitable aspect of online data. For instance, commentary like "please subscribe to my channel" would be commonly found in YouTube videos. However, we believe this challenge could be effectively mitigated with proper data preprocessing. An example from MineCLIP [18] shows that careful heuristic-based preprocessing of the captions can make data clean enough to train multi-modal models (e.g. a simple heuristic focusing on domain-specific vocabulary could remove most off-topic text). More promisingly, one could use language models to filter out off-topic data, either by removing any off-topic comments or determining that some videos should be excluded because not enough of the commentary is helpful. Additionally, even if noise is present, recent ML history shows that–at scale–such "noise" does not prevent learning the "signal": examples of this occurring on internet-scale data include GPT [29], CLIP [30], and VPT [19]. While not a perfect simulation of online caption noise, as described later, we add noise to the synthetic thought data and Thought Cloning works well despite it (Section 3.1).

In the Thought Cloning training framework, agents learn to produce natural language thoughts at each timestep and subsequently condition their actions based on these generated thoughts. This learning process gives rise to a bi-level architecture (Fig. 1). The architecture comprises an Thought Generator responsible for thought generation, and a Action Generator tasked with executing actions based on the thoughts generated by the Thought Generator. While different choices of what to condition the Thought Generator and Action Generator are possible, in this work, for a particular trajectory of length $T$ in the thought dataset we minimize:

$$\min_{\theta_u, \theta_l} \sum_{t=1}^T -\alpha \log \pi_{\theta_u}(th_t|m, \{o_\tau\}_{\tau=1}^t, \{th_\tau\}_{\tau=1}^{t-1}) - \log \pi_{\theta_l}(a_t|m, \{o_\tau\}_{\tau=1}^t, th_t) \quad (1)$$

Here, $\theta_u$ and $\theta_l$ represent the weights for the Thought Generator and Action Generator; $\alpha$ represents the coefficient for Thought Cloning loss; $th$, $o$, $a$, and $m$ denote thought, observation, action, and mission, as previously described. The first part of the loss is the Thought Cloning loss, where the Thought Generator is conditioned on the history of thoughts, observations, and the mission, to predict the thought. That thought is then compared with the ground truth thought in the dataset. The second part is the action loss, where the Action Generator is conditioned on the current thought, the history

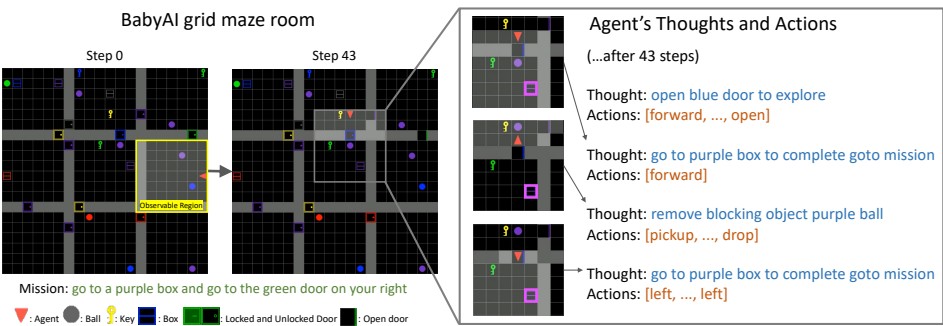

Figure 2: **Left**: A BabyAI [26] environment example. The environment contains various colored items (*ball, key, box, door*). The agent can pick up, drop, and move objects or open and close doors, while locked doors can only be unlocked with color-matched keys. The agent can observe the $7 \times 7$ grid cells in front of it, which can be blocked by walls and closed doors. **Right**: An example from a trained Thought Cloning agent planning and replanning. The mission requires reaching the purple box (highlighted), but a purple ball blocks the way. The agent's thoughts and actions show replanning when encountering the obstacle, removing it, and resuming the previous goal.

of observations, and the mission and predicts the action to do, and we then calculate the loss by comparing the predicted action to the ground truth action in the dataset.

For more complex or large-scale scenarios, the Thought Generator can be implemented with pre-trained Vision-Language Models (VLM) either zero-shot or fine-tuned [31], while the Action Generator can be trained from scratch or adapted from existing language-conditioned controllers in the target domain [6, 16]. In this paper, we base both components on the BabyAI 1.1 model architecture [32], which utilizes a memory-augmented architecture–an LSTM [33]–to address the partial observability challenge. The model also employs FiLM [34] for modality fusion, effectively combining visual and text input. The detailed architecture adopted in this paper can be found in Supplementary Material A. While all models in this paper are trained from scratch, we anticipate that the utilization of pre-trained models in complex domains will be beneficial.

## 3   Experimental Results

### 3.1   Domain and Synthetic Thought Data

This paper employs BabyAI [26], a simulated partially observable 2D gridworld domain. We focus on the most challenging environment, *BossLevel*, in BabyAI. An overview of the domain is shown in Fig. 2 (left). Each BabyAI environment consists of a randomly generated room layout, item configuration, and a mission described in natural language, sampled on an environment distribution. Colored items (*balls, keys, boxs, doors*) and the initial position of the agent are randomly distributed across a $20 \times 20$ grid world containing nine $6 \times 6$ rooms. Missions comprise four possible tasks (*GoTo, PickUp, OpenDoor, PutNextTo*), connected by *then/after* and *and* (with or without ordering constraints). *GoTo* and *PickUp* require agents to go to or pick up an object; *OpenDoor* requires agents to open or unlock a door; *PutNextTo* requires the agent to pick up object A, find object B, and drop A next to B. The mission may implicitly require the agent to open or unlock doors to find the target objects. Relative directional instruction in the mission, e.g., *on your right*, is based on the agent's initial position. An environment is solved when all tasks in the mission are completed. The agent's observation consists of the $7 \times 7$ grid cells in front of the agent, except the agent cannot see through walls (Fig. 2 yellow square). This work features the state-based observations provided by BabyAI [26]. Each grid cell in the $7 \times 7$ observation is represented by three integers: [the item ID, the color ID, and a status code], resulting in a $7 \times 7 \times 3$ observation matrix. The status code is 1 for closed doors and 2 for locked doors, with 0 for open doors and other items. Occluded grid cells are assigned an item ID of 0. The agent's action space includes [left, right, forward, pickup, drop, toggle door (unlock, open, close)].

The key challenges in BabyAI revolve around partial observability, hard-to-explore mazes, complex missions in natural language, and long-horizon planning. The $7 \times 7$ observation field is limited

compared to the $27 \times 27$ maze, and the agent cannot see through walls and closed doors. The maze containing multiple closed rooms is difficult to navigate and explore as the agent needs to find target items across multiple closed (even locked) rooms. The missions are challenging because (1) they are described in natural language and (2) they can consist of multiple tasks, each requiring complicated navigation and actions. Combining all these factors results in a long horizon, with hundreds or even thousands of steps needed to solve a single environment.

One significant advantage of BabyAI is that it provides an Oracle Solver (named BOT in [26]) capable of generating step-by-step solutions for any given environment. This is achieved through hand-coded rules and an internal stack machine to generate plans for solving environments. In our work, we translate the Oracle Solver's internal states into natural language thoughts with pre-defined rules. For example, if the inner logic is to open a red door to explore the room, the translated thought will read, "open red door to explore". This translation process is combined with the generated demonstrations to synthesize the thought dataset with 1 million trajectories. To make the dataset more realistic, noise is added, with a 1% chance of adding a random noisy segment at each timestep, consisting of a random thought and several random actions, with a random length sampled from 1-6. A trajectory with example noise is shown in Supplementary Material C.

## 3.2 Experiment Setup

To verify the effectiveness of *learning to think*, we compare our Thought Cloning (TC) approach to the classic learning algorithm, Behavioral Cloning (BC). BC shares most of its architecture with the Action Generator of TC (Fig. 1), and because it is trained only on action loss, it does not encode thought like the Action Generator of TC. Additionally, since BC has fewer parameters than TC, we introduce an ablation variant called TC w/o Imitating Thought that is trained without the Thought Cloning loss to demonstrate that TC's superiority is not solely due to its larger number of parameters. This variant's architecture is identical to the TC architecture, except for a minor architectural difference where the latent vector from the Thought Generator is directly input to the Thought Generator as thought. This adjustment is necessary because this variant is not trained on the Thought Cloning loss, so we do not have per-word supervision. To train these parameters, we thus need to train them based on how these "thoughts" contribute to the action loss (i.e. they are trained end-to-end to predict actions). If we sampled words (as in Thought Cloning), we could not train these parameters end-to-end because hard sampling of words is non-differentiable, so gradients could not flow backward through this operation. Thus, we make one small change in order to allow the parameters to be trained, which is to pass the logits of the Thought Generator directly into the Action Generator, which is a differentiable operation. We feel this is the closest and fairest control possible to Thought Cloning, allowing virtually the same architecture and the same number of parameters, but not including the Thought Cloning innovation of exploiting human thought data.

Our training setup is based on BabyAI [26, 32]. The training iterates for 8 epochs on the 1 million episode dataset, corresponding to a total of $7 \times 10^8$ training frames. The Thought Cloning loss parameter $\alpha$ (Eq. 1) is set to 2. During training, we employ teacher-forcing [35], which is adopted when decoding thoughts. It conditions the Action Generator on the ground truth thoughts from the dataset. The teacher-forcing ratio linearly decreases from 100% to 0% during the training process. Producing all the main results in the paper took about ten A40 GPUs for one week. More details on training can be found in Supplementary Material A.

In our experiments, the performance of agents is evaluated based on their success rate in *held-out* test environments. Success for an environment is defined as the completion of all specified tasks in the mission. By controlling random seeds, all test environments are unseen during the training process. All experiment results from Sections 3.3, 3.4 and 3.5 are calculated from five independent runs. The success rate results in Section 3.3 are obtained by testing agents on a set of 512 sampled environments. In Section 3.4, agents are tested in a larger set of 1,024 test environments. During the testing phase, the TC agent has identical observations as the BC agent, i.e. it has no extra information.

## 3.3 Imitation Learning

In this section, we show the main performance results of training TC, BC, and TC w/o Imitating Thought. The results illustrate that TC learns faster than BC, where BC requires orders of magnitude more time to achieve a performance similar to TC's early-stage results, and TC ultimately outperforms BC at the end of training (Fig. 3). The outperformance of TC compared to BC at 25%, 50%, 75%, and

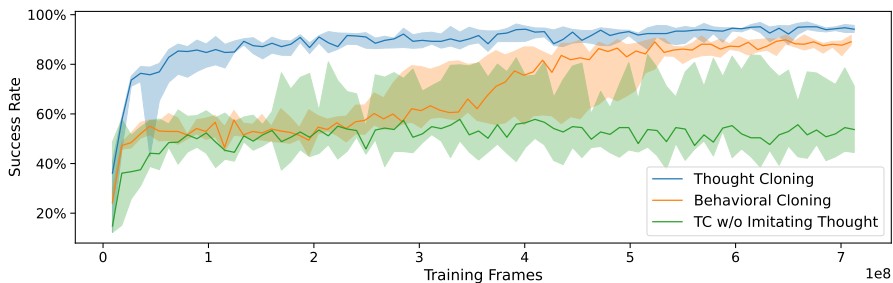

Figure 3: Training progress comparison of Thought Cloning (TC), Behavioral Cloning (BC), and a TC ablation variant without the Thought Cloning loss. The BC architecture is identical to the Action Generator of TC and the TC w/o Imitating Thought has the same architecture as TC, without the TC loss. BC and the ablation variant are trained solely with the action loss (which leads to some minor architectural differences, see Section 3.2.) The error bars are the 95% confidence interval from five runs of experiments. The results indicate that TC learns faster than BC and also outperforms it. Furthermore, the comparison between TC and TC w/o Imitating Thought demonstrates that the superiority of TC is not simply due to having more parameters.

100% of the way through training is statistically significant, as confirmed by the Mann-Whitney U test, with $p = [0.012, 0.008, 0.021, 0.008] < 0.05$. These results support our hypothesis that natural language can help the agent learn to explore and plan.

Another comparison is between TC and an ablation variant TC w/o Imitating Thought that shares the same architecture with TC, but without the Thought Cloning loss in training. The results show that TC also substantially outperforms TC w/o Imitating Thought (Fig. 3). Similar to the previous, the results are statistically significant ($p = [0.008, 0.012, 0.008, 0.008] < 0.05$). The results reveal that TC's superior performance is not solely due to a larger number of parameters than BC, and also supports our argument that learning from human thought boosts an agent's ability to think.

An example of a TC agent planning and replanning is shown in Fig 1 (right). After opening the blue door, the agent discovers the target (a purple box) within its observation and thinks about going to it to complete the task. However, the agent realizes that a purple ball is blocking its path. A smart replan emerges here, with the agent inserting a new plan to remove the ball. The agent achieves this subgoal by picking up the ball in its way, finding an empty space, and then dropping it. After completing this new, necessary, intermediate task, the agent resumes its original mission to go to the purple box and successfully solves the environment. From this example, we can see that by thinking like humans in natural language, the agent demonstrates successful planning and replanning abilities. We also see the interpretability benefits, as it is easy to follow along and understand *why* the agent executes certain actions.

### 3.4 Generalization to Out-of-Distribution Environments

This section compares the generalization abilities of the TC and BC agents by testing them on environments that are increasingly out of distribution. We define the distribution of environments with two difficulty dimensions: Behavioral Difficulty and Cognitive Difficulty. Behavioral Difficulty is based on the length of the action sequence required to solve the environment (provided by Oracle Solver, see Section 3.1). The simplest environments require about 20 steps, while the most challenging environments require more than 500 steps. Cognitive Difficulty reflects the complexity of the mission, with more difficult environments requiring stronger planning abilities to complete complex tasks. The definition of Cognitive Difficulty is adapted from the maxStep parameter in BabyAI environments [26] and is given by (# of {*GoTo, PickUp, OpenDoor*} + 2 × # of {*PutNextTo*} + # of ordering constraints). The PutNextTo task is assigned a higher weight because it involves a combination of picking up, navigating, and dropping, making it the most challenging task among the four. The range of cognitive difficulty spans from 1 (simplest) to 9 (most difficult). In the training distribution, the environments exhibit means and standard deviations of Behavioral and Cognitive Difficulties of $84.2 \pm 68.8$ and $2.7 \pm 1.6$, respectively. In this paper, we define out-of-distribution (OOD) environments as those with a Behavioral Difficulty $> 175$ or a Cognitive Difficulty $\geq 4$, each being approximately more than one standard deviation away from the mean. The furthest OOD environments, with a Behavioral Difficulty greater than 425 or a Cognitive Difficulty of 9, had

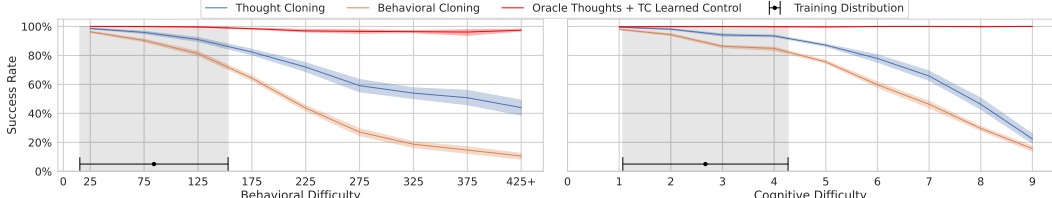

(a) Zero-shot success rates of agents on environments that are increasingly out of distribution.

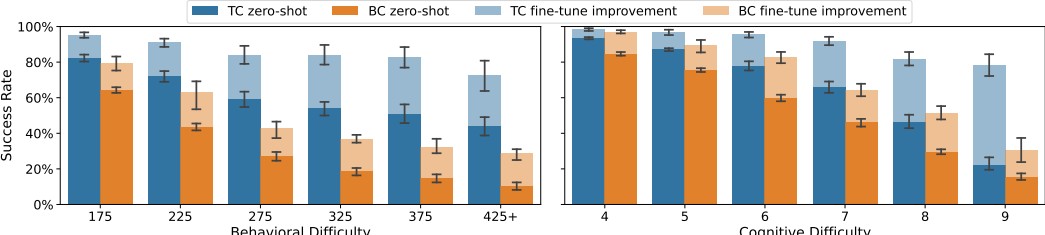

(b) Success rates after fine-tuning agents on out-of-distribution environments

Figure 4: The zero-shot and fine-tuning success rate of Thought Cloning (TC) and Behavioral Cloning (BC) agents on environments that are increasingly out of distribution. Behavioral and Cognitive Difficulties are defined by the length of the solutions to environments and the mission complexity of environments respectively (Section 3.4). The error bars in bar and line plots are the 95% confidence interval from five runs of experiments. (**a**): The gray region indicates the training distribution. The Oracle Thought + TC Learned Control refers to the TC agent with oracle high-level thoughts. The results demonstrate TC generalizes much better than BC. They also illustrate that with a more powerful Thought Generator trained from vast human thought data, the agent should become drastically more capable. (**b**): TC is much better at adapting to novel situations than BC.

less than $5.7 \times 10^{-5}$ and $1.6 \times 10^{-4}$ probability of being sampled during training (calculated with rejection sampling). For testing both in-distribution and out-of-distribution environments, we sample various sets of environments that extend away from the distribution in terms of both Behavioral and Cognitive Difficulty, and then evaluate agents on these sets. For Cognitive Difficulty, we sample sets of environments across the full range of Cognitive Difficulty levels 1-9. For Behavioral Difficulty, we sample sets of environments within intervals of 50 (e.g., 125-175, 175-225, etc.), starting from 25. Environments with a Behavioral Difficulty > 425 are grouped into one set.

First, we test the zero-shot performance of TC and BC agents in OOD environments. The results show that the TC agent substantially outperforms the BC agent with environments being increasingly out of distribution (Fig. 4a), and the results are statistically significant across all testing difficulties (Mann-Whitney U test $p < 0.05$), thereby supporting our hypothesis that language utilization can enhance agents' generalization capabilities. Moreover, we observe that the Oracle Thoughts + TC Learned Control achieves near-optimal performance even on the most challenging environments. This indicates that the current limitation of TC performance lies in high-level thinking. As we scale our approach to leverage internet-sized datasets of human thinking, the high-level thinking capability is expected to improve substantially, thereby enhancing the power of the TC agent.

Next, we investigate how well the agents adapt to new situations by fine-tuning them on OOD environments. We fine-tune the TC and BC agents on the corresponding environments for 15 epochs, with the same settings described in Section 3.2. The results demonstrate that the TC agent is better at adapting to OOD environments (Fig. 4b). The superiority of TC over BC is statistically significant across all testing difficulties, as supported by the Mann-Whitney U test $p < 0.05$, with the exception of Cognitive Difficulty 4, where both methods already achieve near-perfect performance. The results support our argument that language can better assist agents in adapting to novel situations.

## 3.5 AI Safety and Interpretability

The ability to observe the agent's thought process gives our model a high degree of interpretability. To empirically assess the interpretability of TC, we introduce a metric named the Future Action

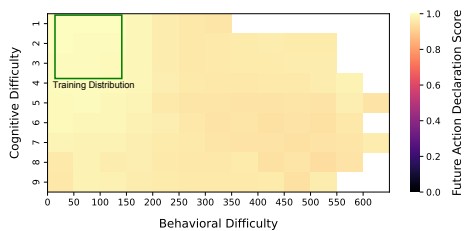

(a) Interpretability of Thought Cloning agents

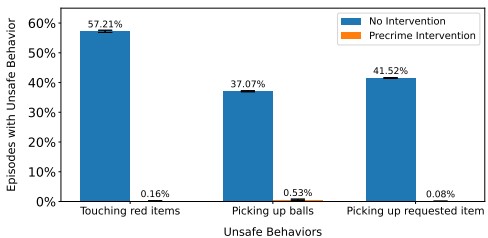

(b) Effectiveness of the Precrime Intervention

Figure 5: (**a**): A heatmap illustrating the Future Action Declaration Score, a metric designed to evaluate the interpretability of Thought Cloning agents (Section 3.5). The $x$ and $y$ axes denote various levels of difficulty. Each cell represents a region of sampled environments, with the color intensity reflecting the mean score. Brighter cells indicate a higher degree of match between the agent's declared thoughts and subsequent actions. The green square denotes the training distribution, while the rest of the regions are out of distribution (Section 3.4). The results illustrate the robust and consistent interpretability of Thought Cloning agents. (**b**): A bar chart demonstrating the effectiveness of the Precrime Intervention mechanism, which is to halt the Thought Cloning agents upon detecting dangerous plans in their thoughts and thus prevent unsafe behaviors. We show three tests ($x$ axis) where (1) touching red items, (2) picking up balls, and (3) picking up requested items were declared unsafe. We report the fraction of episodes where unsafe behaviors occurred ($y$ axis). The error bars are the 95% confidence interval from five runs of experiments. The results show that Precrime Intervention effectively eliminates almost all unsafe behaviors.

Declaration Score. This metric quantifies the fraction of times when an agent, preparing to execute an action other than navigation, declares this impending action in its thoughts beforehand. In the training distribution, TC agents performed exceptionally well (green square in Fig. 5a). Interestingly, TC agents also scored near-perfectly across all out-of-distribution environments (rest of Fig. 5a), demonstrating the robust and consistent interpretability of our model even under novel, out-of-distribution situations, which is an important property for AI safety and interpretability.

Due to its high degree of interpretability, Thought Cloning allows a simple method that can considerably enhance AI safety. We call it Precrime Intervention. In practical settings, an agent might employ dangerous or undesirable strategies to accomplish challenging tasks. However, because Thought Cloning features such strong interpretability, we can simply halt the agent upon detecting dangerous thoughts and thereby prevent the unsafe behavior it was planning to conduct. Additionally and importantly, Precrime Intervention does not require any changes to the weights of the model. If we learn or decide *after* training that a certain behavior is unsafe or undesirable, Precrime Intervention can still prevent it. The same flexibility allows different definitions of what is allowable and unsafe behavior in different settings (e.g. in the presence of adults vs. children or customized to the preferences of different countries with different regulations). To demonstrate this flexibility and test to what extent Precrime Intervention works, we conducted three separate tests, where we declared three different behaviors as unsafe (1) touching any red item, (2) picking up any ball, and (3) picking up the object the agent is being asked to pick up in its mission. The last one is particularly interesting because the agent has a strong prior to want to perform that action, which Precrime Intervention has to combat. We report the fraction of episodes where such unsafe behaviors occurred with and without Precrime Intervention (Fig. 5b). Remarkably, Precrime Intervention almost entirely eliminates all unsafe behaviors, thereby demonstrating the promising potential of TC agents in advancing AI safety.

Moreover, the interpretability of the model also greatly aids in diagnosing problems, thus simplifying the development of more capable and safer AI. This feature actually proved beneficial during the development of this paper. Initially in our development, the TC agent showed promising performance in training, but frequently failed during testing, repetitively oscillating between incorrect thoughts (plans) without actively exploring new ideas. This observation helped us to recognize that, because we had trained with teacher forcing throughout with oracle (i.e. perfect) thoughts, the agent had never practiced having incorrect thoughts, and thus had never practiced recovering from them by trying alternate ideas. Thus, at inference time when it had to generate its own thoughts, which are sometimes incorrect, it did not know how to recover. We then instead tested an immediate transition from teacher-forcing to 100% auto-regressive sampling and training (i.e. from 100% teacher-forcing on

one training step to 0% on the next), but the agent generated too many nonsensical thoughts. Thanks to the model's interpretability, we were able to recognize the situation and try an alternative strategy that worked well, and is the one we report results for in this paper (Section. 3.3): we gradually decay the teacher-forcing rate (fraction) during training, which dramatically improved performance. Supplementary Material D contains more details about this example.

Lastly, TC enables steerability, including helping agents when they are stuck. That is because the actions of TC agents are conditioned on their thoughts, and we can manually inject alternate thoughts to have the agents do what we wish. The TC agent, when provided with Oracle high-level thoughts, is capable of near-perfect performance across almost all environments (Fig. 4a), which provides evidence that steering agents is possible and effective.

## 4 Related Works

### 4.1 Planning in RL with Language

Recent work leverages the reasoning capability, the ability to flexibly combine abstractions, and the interpretability offered by natural language to address high-level planning challenges in real-world domains. We augment this approach by enabling agents to *think in language*, facilitating the capability of agents, AI Safety, and Interpretability. There are two major categories of works in the literature that enable language planning. The first involves Hierarchical RL methods, where the language represents the hierarchy [36–40]. However, the planning space in these works is constrained to a pre-defined subgoal set, limiting their generalization to novel scenarios and preventing them from utilizing the reasoning and powerful commonsense found in pre-trained LLMs [29, 41, 42]. The second category of work involves pre-trained LLMs that generate plans in language for RL systems. Earlier works [16, 17] allow the LLM to predict step-by-step plans for a specific task. However, these works are open-loop methods, as the LLM cannot perceive the environment while acting, and thus cannot adapt and change once things do not go according to plan, which is a crucial capability in complex environments. Some recent approaches have developed closed-loop methods to provide LLMs with dynamic information for planning [17, 43, 44]. While these works show exciting performance in different environments, their closed-loop feedback mechanisms for the LLMs either rely on an oracle from the environment or complicated captioning models. The work most relevant to our vision is PALM-E [31], in which a pre-trained Vision-Language Model is adopted as the planner, allowing it to recognize patterns from observations directly. However, the works mentioned above were not trained on data of humans thinking and acting, meaning they do not benefit from learning from human thought demonstrations how to do things like plan, replan, create high-level goals and the subgoals required to achieve them, and the many other benefits of thinking intelligently during acting.

Apart from embodied domains, many studies utilize LLMs as agents [45] operating in text domains, such as coding [46, 47] or using API tools [48]. Many LLM agents benefit from the reasoning and planning capabilities of LLMs [48, 49]. Some even allow LLMs to replan based on new information [46, 47]. However, none of the methods described above directly learn to think while acting by imitating human thought data. Thus, unlike Thought Cloning, they do not benefit from learning from human thought demonstrations. Within LLM agent research, ReAct [50] is the most relevant to our approach. It prompts LLMs to first generate reasoning and then conditions actions based on such reasoning text. The paper also includes an experiment that fine-tunes LLMs with demonstrations that contain both reasoning and actions. However, obtaining such demonstrations is challenging, as highlighted in the ReAct paper [50], and the paper does not provide insights on overcoming this data bottleneck to scale the method.

### 4.2 Learning from Dataset Aligning Action and Language

Several studies have recognized the value of datasets that align action with language. DIAL [51] employs such a dataset to train language-conditioned agents with Behavioral Cloning, achieving impressive results in real-world robotic tasks. However, it is limited by a pre-defined instruction set. Another work, (SL)$^3$ [52], generates a hierarchical dataset for agents to learn from, demonstrating superiority in a challenging 3D simulation domain, but has the drawback discussed in the previous section of being open-loop. Finally, in the study most similar to our own, Hu et al. [53] collected a dataset from two human players collaborating on an RTS game. However, the agent in [53] is not language conditioned, which limits its potential to learn to do any task (e.g. arbitrary tasks

requested of it in natural language). Similarly, a work concurrent to ours constructed a dataset with BabyAI oracle plans [54]. However, their architecture, unlike ours, is not compatible with most pre-trained models, making ours more able to harness new, powerful foundation models to tackle increasingly complex challenges. Additionally, although previously mentioned two methods [53, 54] employ learning frameworks similar to ours, they do not explore the full potential of learning from datasets that align action with language, particularly in terms of the resulting benefits in terms of generalization, AI Safety, and Interpretability.

## 5   Discussion and Conclusion

Our research findings are focused on two main areas. First, our Thought Cloning (TC) agent demonstrated superior performance compared to Behavioral Cloning (BC), effectively showcasing its capabilities in generalization, exploration, planning, replanning, and adaptation to various situations. Second, we presented empirical evidence underscoring the benefits Thought Cloning provides in AI Safety and Interpretability. The robust interpretability of the TC agent not only help developers in diagnosing AI systems, but also contributes to AI safety, as evidenced by mechanisms such as Precrime Intervention. Our empirical results on steerability further spotlight the potential of TC agents in effectively collaborating with humans to tackle complex tasks.

We utilized a synthetic dataset and trained a model from scratch as a proof of concept. However, the full vision for the TC framework will be when Thought Cloning agents are trained on internet-scale datasets of humans thinking out loud while acting, such as YouTube videos and their narration [19, 18] (whether in closed caption transcripts or directly from audio). Consider the prospect of an agent that has both learned to think and act like humans in a huge variety of settings. Much like the thoughts of human children are guided by teachers, our agents could become skilled at planning, replanning, reasoning, and explaining their thinking to us (either via their outputs or because we have the unique ability to observe the thoughts in their minds). The vision for utilizing internet-scale datasets is also supported by the empirical results in Section 3.4, which suggest that the current bottleneck in agent capability is its high-level thinking, a skill that could be enhanced by scaling to vast online data. Additionally, we trained all models from scratch in this paper. In more complex domains, we anticipate that the utilization of pre-trained Transformer-based models could be beneficial.

Moreover, Thought Cloning can also improve Foundation Models (FMs) by enabling a separate "thought channel" where they can output thoughts that get fed back in when they are planning and answering. Recent studies indicate that FMs can benefit from contexts they self-generate, such as reasoning steps [11, 50]. Thought Cloning can amplify this advantage by training FMs with Thought Cloning data which includes not only the answers or solutions (action data) but also the reasoning behind them (thought data). For example, a human programmer needs the ability to think when developing software, but in a way where those thoughts are not part of the final output. However, current LLMs are not trained this way. In addition to performance gains, adding such a thought channel to FMs allows all of the AI Safety and Interpretability advantages of Thought Cloning, including Precrime Intervention, to be added to the world-changing FM technology.

Of course, there are also risks associated with such agents. As occurs with LLM pretraining and other forms of Behavioral Cloning, Thought Cloning could inadvertently inherit undesirable human flaws, such as speaking falsehoods, providing false yet persuasive rationalizations, or being biased. Alignment techniques are being constantly improved to address these challenges [55]. However, even improving AI agent safety up to the level of a (flawed) human would be a major advance, even if the resulting system is not perfect. Additionally, a distinguishing feature of Thought Cloning is it provides the ability to interpret and prevent these flaws from culminating into actions, making TC a more favorable approach in this regard. (See the experimental analyses on the effectiveness of Precrime Intervention Section 3.5). Our intent was to inspire researchers to advance this method and or test it in real-world scenarios, but the method itself should not be considered as a comprehensive safety solution for real-world systems.

In conclusion, this paper has introduced Thought Cloning, where agents not only simply learn to act from human demonstrations, as in Behavioral Cloning, but also *learn to think* from demonstrations where humans think out loud while acting. Through Thought Cloning, we have illustrated how an agent can become more capable, interpretable, and safe by *imitating human thinking*. This work facilitates the training of increasingly powerful agents and opens up numerous avenues for future scientific investigation in Artificial General Intelligence, AI Safety, and Interpretability.

## Acknowledgments and Disclosure of Funding

This work was supported by the Vector Institute, the Canada CIFAR AI Chairs program, grants from Schmidt Futures and Open Philanthropy, an NSERC Discovery Grant, and a generous donation from Rafael Cosman. We thank Aaron Dharna, Ben Norman, and Jenny Zhang (sorted alphabetically) in our lab at the University of British Columbia for insightful discussions, and Michiel van de Panne for helpful commentary.

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

# Supplementary Material

## A   Architecture and Training Details

For full transparency, replicability, and to facilitate future research building on our work, we are releasing the source code, model weights, and the dataset used in our experiments[2]. Additionally, we provide key details necessary for the evaluation and replication of our work in this supplementary information. The architectural details of the Thought Cloning models adopted in this paper are shown in Fig. 6. As in [32], all missions and thoughts are encoded with Gated Linear Units (GLUs), with separate encoders employed for the missions and thoughts respectively. After the encoding process, we apply an attention mechanism [56] to dynamically weight the importance of different parts of the text encoding, based on the state history. The observation is encoded with a Convolutional Neural Network (CNN) and a Bag-of-Words [57] encoding approach. In the Thought Generator, a Transformer encoder [56, 58] is adopted to embed the thought history and mission, with the thought history as the query and the mission as the key and value. This Transformer encoder consists of two layers, each with two heads. The Action Generator is identical to the Behavior Cloning Baseline, except with the additional encoding of thoughts. Key architectural parameters, such as memory size and embedding size, are consistent with the baseline in [32], as shown in Table 1.

The pseudocode for Thought Cloning (TC) training framework is shown in Algorithm 1. In the loss function, we follow [32] by including an entropy term for actions. The Adam optimizer [59] is adopted to train TC and TC variant, with a batch size of 180 and a learning rate of $5e^{-4}$. Similar to the setting in baseline [32, 26], we train BC with a batch size of 296 and a learning rate $5e^{-5}$. The learning rate schedule begins with a warm-up phase of $5T$ training steps, where $T = 51200$, linearly increasing from $1e^{-4}$ to $5e^{-4}$ for every $T$ step, and then decaying by 50% at $120T$ training steps, similar to the practices in [26, 60]. The teacher-forcing ratio linearly decreases from 100% to 0% from the $10T$ training step to the end for every $T$ step. In line 5 of Algorithm 1, the input thought could be the ground truth from the dataset $(th_t)$ or the generated thought from the Thought Generator $(\hat{th}_t)$, depending on with or without teacher forcing. For training efficiency, Backpropagation Through Time was truncated at 20 steps in TC. The mix precision in PyTorch is also adopted during training, which speeds up training without sacrificing much performance [61]. In fine-tuning experiments, due to the increased difficulty of the levels and longer steps requiring more memory, we reduced the batch size from 180 to 40 and trained with an auto-regressive strategy. Detailed hyperparameter settings are shown in Table 1.

Table 1: Hyperparameter Settings

| Hyperparameter | Value |
|---|---|
| Adam $\beta_1$ | 0.9 |
| Adam $\beta_2$ | 0.99 |
| Adam $\epsilon$ | $10^{-5}$ |
| Entropy Coefficient | 0.01 |
| Image Embedding Dimension | 128 |
| Text Embedding Dimension | 256 |
| Memory Dimension | 2048 |

## B   Ablation Study: BC with the same parameter count or data as TC

We designed the control TC w/o Imitating Thought (Section 3.2) to address the concern that TC outperforms BC because TC has more parameters. Although TC w/o Imitating Thought allows the control to have the same number of parameters (and architecture) as Thought Cloning, for completeness, we try to address the concern in another way. Instead of holding the architecture the same, we create a BC control with a more canonical BC architecture, but with the same number of

---

[2]`https://github.com/ShengranHu/Thought-Cloning`

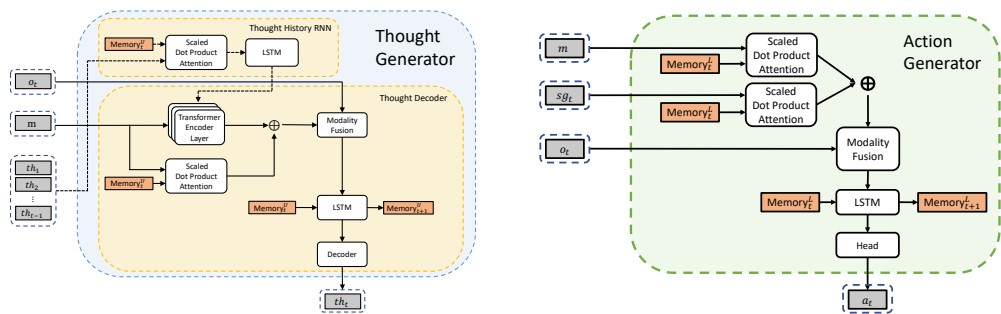

Figure 6: Detailed architecture for Thought Cloning (TC) agent adopted in this paper. At each timestep $t$, the inputs to the TC agent include a natural language-defined mission $m$, and an observation $o_t$. All preceding thoughts $\{th_\tau\}_{\tau=1}^{t-1}$ is embedded with an LSTM in the Thought Generator. The generated thought $th_t$ from the Thought Generator will be the input to the Action Generator and an action $a_t$ is predicted by the Action Generator. (**Left**): The Thought Generator. We employ an LSTM [33] to embed the thought history and a transformer encoder to process both the mission and thought history. The text input is then fused with the visual observation input using FiLM [34]. (**Right**): The Action Generator is largely similar to the BabyAI agent [26], with the primary difference being the additional embedding of the thought generated by the Thought Generator.

---

**Algorithm 1** Thought Cloning

1: **Input:** thought dataset $\mathcal{D} = \{D_i\}_{i=1}^N$, where each $D_i = (m, \{(o_t, th_t, a_t)\}_{t=1}^T)$, Thought Generator $\pi_{\theta_u}(th|o, m, \{history\_th\})$, Action Generator $\pi_{\theta_l}(a|o, m, th)$
2: **while** training **do**
3:     **for** each $D_i = (m, \{(o_t, th_t, a_t)\}_{t=1}^T)$ in $\mathcal{D}$ **do**
4:         **for** each $(o_t, th_t, a_t)$ in $D_i$ **do**
5:             Generate thought sentence $\hat{th}_t = \pi_{\theta_u}(\cdot|m, \{o_\tau\}_{\tau=1}^t, \{th_\tau\}_{\tau=1}^{t-1})$
6:             Predict action probability distribution $\hat{a}_t = \pi_{\theta_l}(\cdot|m, \{o_\tau\}_{\tau=1}^t, \hat{th}_t)$
7:             Compute the loss: $\mathcal{L}(\theta_u, \theta_l) = \mathcal{L}_{CE}(a_t, \hat{a}_t) + \alpha \mathcal{L}_{CE}(th_t, \hat{th}_t) - \beta H(\hat{a}_t)$
8:             Update the policy network parameters $\theta_u, \theta_l$ by minimizing $\mathcal{L}(\theta_u, \theta_l)$
9:         **end for**
10:     **end for**
11: **end while**

---

parameters as TC. Results show it does not perform nearly as well as TC (Fig. 7 and Table 2: "Pure BC architecture (and matched num parameters)").

Also, one might argue that TC gets more data (one set of actions and one set of words per episode), and thus that a proper BC control is to give BC twice as much data. A counter is that such a control is unnecessary because noticing and harnessing this additional (often free) and heretofore ignored data stream is a central contribution of this paper, so showing that using this data improves things is the main comparison to be made. However, for completeness, we ran an experiment giving BC twice as much data: results show BC with twice as much data is still far slower to learn and has significantly worse performance at convergence (Fig. 7 and Table 2: "BC w/ 2x data"). Additionally, a prior work Think Before You Act [54] has a similar number of episodes of actions and language data in the same domain, and their model has more parameters than ours. Results show that TC also outperforms that method (Table 2).

Finally, to further confirm that the superiority of TC did not come from both more data and parameters, an ablation study was conducted on a pure BC architecture with the same number of parameters as TC but also with 2x data. The, results show that it also underperformed TC (Fig. 7 and Table 2: "Pure BC architecture (and matched num parameters), 2x data"). All of the aforementioned results highlight the effectiveness of the TC framework and the importance of the thought data.

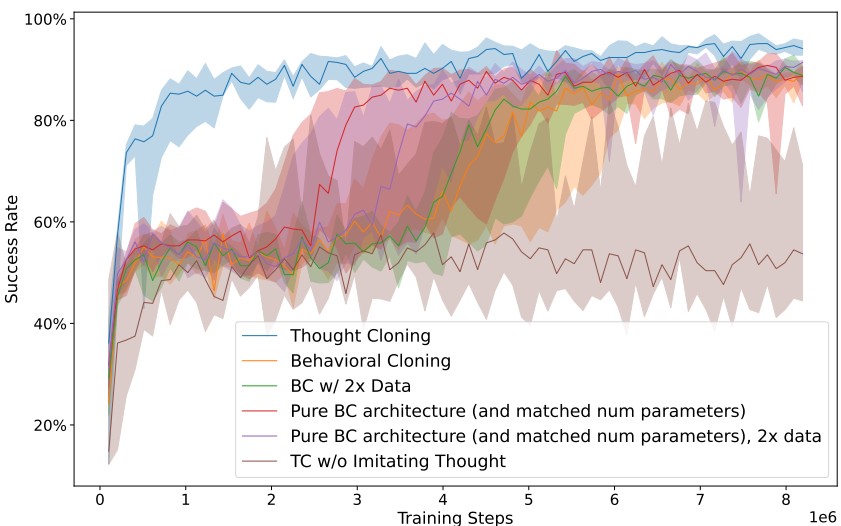

Figure 7: Training progress comparison of Thought Cloning (TC), Behavioral Cloning (BC), and other ablation variants. The BC architecture is identical to the Action Generator of TC and the TC w/o Imitating Thought has the same architecture as TC, but without the TC loss (and has one minor architectural change that is required since it does not have this loss; see response to reviewer gnSR). BC w/ 2x Data trains BC with 2x more episodes worth of actions than TC and BC. Pure BC architecture (and matched num parameters) is to augment BC architecture to have matched parameters with TC. Pure BC architecture (and matched num parameters), 2x data trains BC with both 2x episodes of actions and a BC architecture scaled up to match the number of parameters of TC. See Table 2 in this document for more details on the number of parameters and amount of data for each treatment. The error bars are the 95% confidence intervals from five runs of experiments. The results show that TC learns dramatically faster and has significantly higher performance at convergence compared with BC and all ablation variants, which indicates the advantages of TC are not solely due to additional training data or model capacity.

## C    Synthetic Human Thought Dataset

Fig. 8 presents an example trajectory. We translate the inner state of the BabyAI Oracle Solver (called "Bot" in [26]) into natural language thoughts. These thoughts outline the current plan for task completion and also describe the underlying intentions behind these plans, as the same low-level plan can serve different stated high-level purposes. For instance, the plan could be to *"open the red door"* with the intention of *"completing the open mission"* or *"exploring"*. The segments with inserted noise are marked in red in Fig. 8.

## D    Example on Diagnosing Agents by Observing Thoughts

In this section, we provide an example of one time when we were able to diagnose Thought Cloning (TC) agents by observing their thoughts during the development phase of this paper. In the early stages of development, we trained the TC agent with a constant teacher-forcing strategy. We observed that during testing, the agents often got stuck persisting with incorrect thoughts and did not actively explore new ideas. For instance, in the top right example in Fig. 9, after $t$=53, the agent persistently attempted to implement the incorrect thought *"pickup blue box to complete putnext mission"* until it reached the maximum step limit, without seeking new ideas. This observation led us to realize that, as we exclusively trained the agent with oracle thoughts via a teacher-forcing strategy, the agent had never practiced dealing with incorrect thoughts and consequently had not learned to recover from them by trying alternative ideas. Subsequently before this realization, we had attempted to transition directly to auto-regressive training after the teacher-forcing training stage. However, the agent then started to generate nonsensical thoughts. The trajectory in Fig. 9 (left) shows nonsensical thoughts such as *open blue* ($t$=24) and *pickup door door* ($t$=75) being generated when a constant auto-regressive strategy is applied. Because of the realization from being able to observe the agent's

**mission: put a key next to the blue ball and pick up the green key**

t=1-5: Thought "pickup a key to complete PutNext mission", Action [forward, right, ... , forward]
t=6: Thought "pickup blue key to complete PutNext mission", Action [pickup]
t=7-12: Thought "explore unseen area", Action [right, right, ... , right]
t=13-14: Thought "open green door to explore", Action [left, left, ... , toggle]
t=15-22: Thought "explore unseen area", Action [forward, forward, ... , left]
t=23-28: Thought "open red door to explore", Action [forward, forward, ... , toggle]
t=29: Thought "explore unseen area", Action [forward]
t=30-38: Thought "drop blue key to complete PutNext mission", Action [forward, forward, ... , drop]
t=39-44: Thought "explore unseen area", Action [right, right, ... , left]
t=45-48: Thought "open green door to explore", Action [left, left, ... , toggle]
t=49-56: Thought "explore unseen area", Action [forward, forward, ... , left]
t=57-69: Thought "open green door to explore", Action [forward, forward, ... , toggle]
t=70-77: Thought "explore unseen area", Action [forward, forward, ... , right]
t=78-88: Thought "open grey door to explore", Action [forward, forward, ... , toggle]
t=89-95: Thought "explore unseen area", Action [forward, forward, ... , right]
t=96-101: Thought "open purple door to explore", Action [forward, forward, ... , toggle]
t=102-110: Thought "explore unseen area", Action [forward, forward, ... , right]
t=111-122: Thought "open red door to explore", Action [forward, forward, ... , toggle]
t=123-130: Thought "explore unseen area", Action [forward, forward, ... , right]
t=131-134: Thought "open grey door to explore", Action [right, right, ... , forward]
t=135-137: Thought "pickup green box to complete PutNext mission", Action [left, left, left]
t=138-204: Thought "open grey door to explore", Action [right, right, ... , toggle]
t=205: Thought "explore unseen area", Action [forward]
t=206-213: Thought "pickup green key to complete pickup mission", Action [forward,...,pickup]

Figure 8: Example trajectories of the synthetic human thought dataset. The inserted noisy segments are highlighted in red.

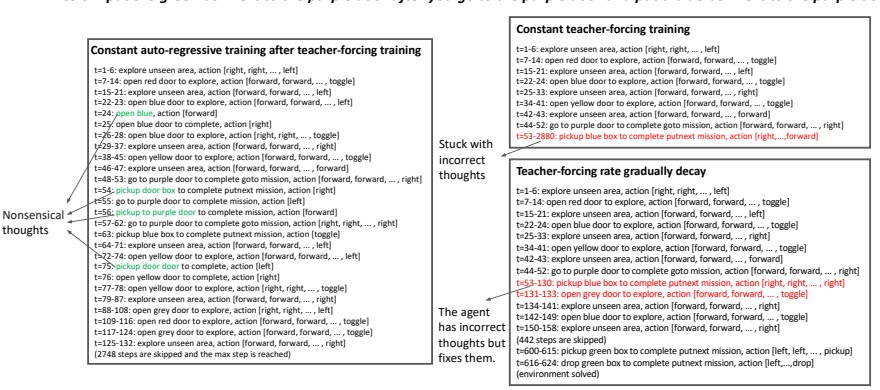

Figure 9: Example trajectories of agents trained with different strategies. **Constant teacher-forcing training** refers to exclusively training with the teacher-forcing strategy. In this scenario, the agent does not learn to recover from incorrect thoughts. Once it adopts an incorrect thought, it continues to follow this thought for thousands of time-steps until it reaches the maximum step count (top right from $t$=53 to $t$=2880). **Constant auto-regressive training after teacher-forcing training** implies directly transitioning to auto-regressive training following an initial phase of teacher-forcing training. In this case, agents begin to generate nonsensical thoughts, as shown on the left, such as *open blue* at $t$=24 (left) and *pickup door door* at $t$=75 (left). **Gradual decay of teacher-forcing rate** involves gradually reducing the ratio of teacher-forcing during training. This strategy is adopted in the final version of Thought Cloning. In this setting, the agent might generate some incorrect thoughts as shown at $t$=53 (bottom right), but it can recover from these errors to explore new ideas, as evidenced at $t$=131 (bottom right).

Table 2: Success rates of TC, BC, ablation variants, and the related work Think Before You Act [54] on BabyAI *BossLevel*. See the descriptions of ablation variants in the Fig. 7 Caption. The Success Rate is the mean and standard deviation from 5 experiments (except Think Before You Act, which did not include the number of experiments in their paper). The num (nubmer of) parameters of Think Before You Act is estimated from the number of parameters of the Huggingface GPT-2 model [62]. The results show TC outperforms all other methods, illustrating the superiority of TC, and supporting our argument that the advantages of TC are not solely due to additional training data or model capacity.

| Method | Num Parameters | Data | Success Rate (%) |
|---|---|---|---|
| Behavioral Cloning | 20.6M | 1M Episodes Actions | $91.2 \pm 0.9$ |
| Thought Cloning w/o Imitating Thought | 82.5M | 1M Episodes Actions | $65.5 \pm 12.4$ |
| BC w/ 2x Data | 20.6M | 2M Episodes Actions | $91.4 \pm 1.7$ |
| Pure BC architecture (and matched num parameters) | 83.9M | 1M Episodes Actions | $91.9 \pm 2.1$ |
| Pure BC architecture (and matched num parameters), 2x data | 83.9M | 2M Episodes Actions | $92.7 \pm 1.0$ |
| Think Before You Act [54] | 124M (estimated) | 1M Episodes Actions + Language | $85.2 \pm 0.5$ |
| **Thought Cloning** | 82.5M | 1M Episodes Actions + Language | $\mathbf{96.2 \pm 0.8}$ |

thoughts, we adopted a gradual decay schedule for teacher-forcing rates during training. As shown in Fig. 9 (bottom right), the agent in this setting was able to explore new ideas after failing on an incorrect thought, and it rarely generate nonsensical thoughts. For example, the agent generates an incorrect thought at $t$=53, but it can recover from these errors to explore new ideas, e.g. *open grey door to explore*. Because we can observe the TC agents *thinking out loud*, we are able to identify the issue and improve the agent's performance. Without this visibility into the agent's thoughts, simply observing their actions would have made it much harder to pinpoint the underlying problems.

