# OpenReview forum: "Thought Cloning: Learning to Think while Acting by Imitating Human Thinking"
_NeurIPS.cc/2023/Conference — NeurIPS 2023 spotlight_

### Official Review · Reviewer_HBeY · 2023-06-28

**Soundness:** 4 excellent
**Presentation:** 4 excellent
**Contribution:** 3 good
**Rating:** 9
**Confidence:** 4

**Summary:**

This paper develops a method for "thought cloning", which involves imitating a human's thought process during task performance. The authors apply this to a partially observable 2D grid world domain. They show that thought cloning yields superior performance compared to behavior cloning.

**Strengths:**

- The approach is innovative, and could be highly impactful once it is scaled up.

- The paper is clearly written.

- The methods are sound, as far as I can tell.

- The application to AI safety and interpretability is interesting and increases the significance of the contribution.

- The paper makes a strong empirical case that their approach works, at least for gridworlds.

- I liked the cognitive science motivation, even if I might quibble with some of the arguments.

**Weaknesses:**

- It would have made a stronger contribution if the authors could show that this method scales up to more complex domains, though I appreciate that introduces many new difficulties and they wanted to show that the approach works on simpler domains first.

- I think the authors are a bit loose with their arguments about human cognition. It is hotly debate to what extent cognition depends on language in a strong way. It also important to note that one can endorse a "language of thought" hypothesis about high-level cognition without endorsing the hypothesis that this corresponds to natural language. In any case, I appreciate that these points have little bearing on the substance of this paper.

- Eq. 1 could benefit from more explanation.

- Please state what error bars show in figures.

UPDATE: the authors have addressed my comments.

**Questions:**

I think the paper is basically good as is. If it were a longer format, I would ask about how the authors would scale up their approach (as they discuss briefly in the paper).

UPDATE: the authors have addressed my comments.

**Limitations:**

Yes, the authors have adequately addressed the limitations.

---

> ### Author Rebuttal · Authors · 2023-08-09
>
> Thank you for your comprehensive review of our paper and your acknowledgment of the strengths of our work, including the novelty and soundness of the method, the strong empirical case, and the contribution to AI Safety and Interpretability. We appreciate the depth of your feedback and are pleased to note your positive assessment. We address your concerns and suggestions below.
>
> ***“It would have made a stronger contribution if the authors could show that this method scales up to more complex domains, though I appreciate that introduces many new difficulties and they wanted to show that the approach works on simpler domains first.”***
>
> We think this is a great direction for future work. As you say, that would introduce additional difficulties and we wanted to show (and study) the approach on a simpler (and easier to analyze) domain first. We believe the best domain to do the science for this paper introducing the method was BabyAI. It is in fact a quite challenging domain, where there are complicated language-described tasks and long-horizon action controls, but it also has other key attributes, such as the ability to generate synthetic thought data and being easy to analyze. We believe the core principles of Thought Cloning hold promise in more challenging domains, and that much interesting future work exists to see how it performs in such domains.
>
> ***“I think the authors are a bit loose with their arguments about human cognition. It is hotly debate to what extent cognition depends on language in a strong way. It also important to note that one can endorse a "language of thought" hypothesis about high-level cognition without endorsing the hypothesis that this corresponds to natural language. In any case, I appreciate that these points have little bearing on the substance of this paper.”***
>
> Thank you for your comments. Please see the general reviewer response, where this is addressed.
>
> ***“Eq. 1 could benefit from more explanation.”***
>
> Thank you for your comment. We will add the following text in L104 in the manuscript.
>
> “The first part of the loss is the Thought Cloning loss, where the Upper-level Component is conditioned on the history of thoughts and observations, and the mission, to predict the thought. That thought is then compared with the ground truth thought in the dataset. The second part is the action loss, where the Lower-level Component is conditioned on the current thought, the history of observations, and the mission to predict the action to do, and we then calculate the loss by comparing the predicted action to the ground truth action in the dataset. ”
>
> ***“Please state what error bars show in figures.”***
>
> Thank you for pointing this out. The error bars are the 95% confidence interval from five runs of experiments. We will add the explanation in the related figure captions.
>
> Thank you once again for your review and positive comments. Your score is already high and we appreciate that, but we wonder if you feel the paper is stronger having read our responses if you might consider increasing it further to help its chance of being published and shared with the ML community. We deeply thank you for your consideration.

---

> > ### Comment · Reviewer_HBeY · 2023-08-14
> > **response to rebuttal**
> >
> > I thank the authors for addressing my comments. I believe the paper should be accepted, and I'm raising my score to 9.

---

> > > ### Author Response · Authors · 2023-08-15
> > > **Thank you**
> > >
> > > Thank you very much. We deeply appreciate your time, insightful review, thoughtful consideration, and for increasing your score. We hope the paper is published and, if it is, look forward to sharing it with the NeurIPS community.

---

### Official Review · Reviewer_9iZD · 2023-07-04

**Soundness:** 3 good
**Presentation:** 4 excellent
**Contribution:** 3 good
**Rating:** 7
**Confidence:** 4

**Summary:**

This paper provides Thought Cloning (TC), an imitation learning method that clones not only behaviors but also thoughts. Here, thoughts are descriptive texts for each behaviors. The basic idea is that language can help agents to better plan their actions and adapt to a new environment. More specifically, the TC agent consists of two components: the Upper-Level and Lower-Level Components. The Upper-Level Component learns to generate a thought conditioned on an observation, a mission, and a history of thoughts. The Lower-Level Component learns to generate an action conditioned on an observation, a mission, and the generated thought. This paper demonstrated the effectiveness of TC on BabyAI, since synthetic thoughts can be easily generated on BabyAI. The experiments showed that TC can be trained much faster than Behavioral Cloning (BC), and TC can better generalize to out-of-distribution environment than BC.

**Strengths:**

This paper has some strong points as follows.

- First of all, this paper is very well written and organized.

- The idea of generating actions conditioned on thoughts (descriptive texts for actions) is simple but effective.

- Thought Cloning (TC) seems have some advantages compared to Behavioral Cloning (BC). First, TC is trained faster than BC. Second, TC may improve the ability to generalize to out-of-distribution environments. Third, TC also enhance the interpretability, since human can see descriptive texts for each actions.

**Weaknesses:**

This paper has some weak points as follows.

- One of my main concerns is how can we effectively collect thoughts for actions to train TC agents. For the purpose of demonstration, this paper trained the TC agent on BabyAI where thoughts for actions are synthetically generated. However, in real world scenarios, it is not easy to collect (high quality) thoughts for actions. Even though the authors mentioned that Youtube videos and vision-language models (VLM) can be used for collecting thoughts for actions, such thoughts would be very noisy compared to synthetic thoughts on BabyAI. Accordingly, the training efficiency and the performance may be deteriorated.

- This paper implemented the TC agent mainly based on LSTM. I am not sure that LSTM can effectively generate thoughts conditions on a history of thoughts. It would be interesting to show the performance of a Transformer-based TC agent.

- The main experiments of this paper was performed on only one environment, BabyAI. It would be better to provide experiment results on other environments.


**Questions:**

Q1. How can we effectively collect thoughts for actions to train TC agents?

Q2. How much robust is the TC agent with regard to the quality of training thoughts data? What is the effect, if we control the quality of synthetic thoughts on BabyAI?

Q3. Is LSTM proper as a base model for TC agents? If we use Transformers instead of LSTMs, is the performance improved or not?

Q4. How well does the TC agent perform on other environments in addition to BabyAI?

**Limitations:**

- The effectiveness of TC agents seems to be highly dependent on the quality of thoughts for actions. It would be better to provide a way of efficiently collecting high-quality thought data. Also, it would be interesting to show how much robust the TC agent with regard to the level of quality of thoughts.

- It would be better to provide more results on other environments.

---

> ### Author Rebuttal · Authors · 2023-08-09
>
> Thank you for your thoughtful review and feedback on our manuscript. We are delighted that you consider our idea simple yet effective, and recognize our contribution to the capabilities and interpretability of agents. Below we address each of your concerns and questions.
>
> ***“How can we effectively collect thoughts for actions to train TC agents?”***
>
> Thank you for your comments. Please see the general reviewer response, where this is addressed.
>
> ***“Even though the authors mentioned that Youtube videos and vision-language models (VLM) can be used for collecting thoughts for actions, such thoughts would be very noisy compared to synthetic thoughts on BabyAI.”***
>
> We recognize the noise in internet data and that it potentially could be an issue. However, we believe this challenge could be effectively mitigated with proper data processing. An example from MineCLIP [7] shows that careful heuristic-based processing of the captions could make data clean enough to train multi-modal models (e.g. a simple heuristic focusing on domain-specific vocabulary could remove most off-topic text like “welcome to my channel”). More promisingly, one could use language models to filter out off-topic data (they are quite capable and will only get better), either by removing any off-topic comments or determining that some videos should be excluded because not enough of the commentary is helpful. Additionally, even if noise is present, recent ML history shows that–at scale–such “noise” does not prevent learning the “signal”: examples of this occurring on internet-scale data include GPT, CLIP, and VPT (all of which one might have argued ahead of time would not work due to training on noisy data, but in fact work very well despite noise). We view these as fascinating topics of future research inspired by our paper and will say so in the revised paper.
>
> ***“How much robust is the TC agent with regard to the quality of training thoughts data? What is the effect, if we control the quality of synthetic thoughts on BabyAI?”***
>
> We added noise to our training data and found TC is robust to it (in fact, it helps because it trains the agent how to get back on track if it makes a mistake in its thinking). Currently, about 3.5% of data is noisy data. If you like, we would be happy to do an experimental sweep of different fractions of noise. Given how long it takes to perform TC runs, it was not possible to do that sweep during the short amount of time in the rebuttal window. If you request it, we could also do tests with other types of noise, including adding phrases like “please subscribe to my channel” and additional tests using LLMs to filter out such off-topic phrases, but we believe such a substantial additional set of experiments is better left to future work where there is space to properly document it..
>
> ***“Is LSTM proper as a base model for TC agents? If we use Transformers instead of LSTMs, is the performance improved or not?”***
>
> We recognize the superiority of the Transformers architecture in recent work. If it comes to real-world scenarios with large-scale language or multimodal tasks, a Transformer would likely be a more powerful model and would be a natural choice. However, in our test domain (BabyAI), it is still a middle-scale problem compared to the internet-scale dataset where Transformers truly shine. Thus, the data-hungry nature of Transformers could make it not the most powerful architecture, or at least not necessary, in BabyAI. One example is from Think Before You Act (Mezghani et al. 2023), where they use GPT-2 like Transformers in the BabyAI environment, but their performance (85.2% Success Rate on BossLevel) is outperformed by BabyAI1.1 (Hui et al. 2020), an LSTM-based architecture baseline (90.4% Success Rate on BossLevel), which has fewer parameters than Think Before You Act. We will mention the value of testing with transformers in future work.
>
> ***“How well does the TC agent perform on other environments in addition to BabyAI?”***
>
> We think this is a great direction for future work. However, we believe the best domain to do the science for this paper introducing the method was BabyAI. It is in fact a quite challenging domain, where there are complicated language-described tasks and long-horizon action controls, but it also has other key attributes, such as the ability to generate synthetic thought data. We believe the core principles of Thought Cloning hold promise in more challenging domains, and that much interesting future work exists to see how it performs in such domains.
>
> Thank you once again for your review and positive comments. Your score is already high and we appreciate that, but we wonder if you feel the paper is stronger having read our responses if you might consider increasing it further to help its chance of being published and shared with the ML community. We deeply thank you for your consideration.

---

> > ### Comment · Reviewer_9iZD · 2023-08-20
> >
> > Thank you for providing a thoughtful author response. I have carefully read the response. I think that my major concern (e.g., collecting high-quality thoughts for actions to train TC agents) has been largely addressed. Even though TC was evaluated only on BabiAI, I think that this paper is a compelling early work that can promote interesting follow-up research in the community. Therefore, it would be beneficial for this paper to be published in NeurIPS.

---

### Official Review · Reviewer_QGuE · 2023-07-06

**Soundness:** 2 fair
**Presentation:** 3 good
**Contribution:** 1 poor
**Rating:** 5
**Confidence:** 3

**Summary:**

The paper studies the problem of imitation learning and attempt to improve existing IL algorithms by training the agent to think like the expert that is being imitated. Authors propose Thought Cloning - an extension to behavior cloning that seeks to imitate thoughts expressed in natural language. They evaluate the proposed method on thought-augmented data from solving tasks in the BabyAI gridworld. Authors compare the proposed algorithm to standard behavior cloning and observe faster training and better generalization. The paper also contains extended discussion about the benefits of thought cloning with respect to AI safety.

**Strengths:**

1. The Thought Cloning method appears easonable and exploits the paper's high level idea in a natural / intuitive fashion.
2. The paper contains a reasonable evaluation of the proposed idea by augmenting popular environments with expert's "thoughts". Typical caveats are addressed with a reasonable a reasonable ablation analysis.
3. The paper is well-written and easy to follow. The general presentation quality is high.

**Weaknesses:**

### Concern 1: lack of comparison with prior work

There are several previous papers that also attempt to teach an agent to think using natural langauge. Authors address planning in natural langauge in section 4.1, but there is a line of work that is not covered by planning. From a surface-level analysis of existing work, there are papers that

* Improve reinforcement learning by performing self-feedback - Madaan et al, Self-Refine: Iterative Refinement with Self-Feedback
* Ask a language model to reflect on it's performance in natural language, improving learning - Shin et al, Reflexion: an autonomous agent with dynamic memory and self-reflection
* Produce natural langauge explanation - parts of Wang et al, Describe, Explain, Plan and Select: Interactive Planning with Large Language Models Enables Open-World Multi-Task Agents

Since this is only a surface-level analysis by a non-expert (me), there could be more related works that study this problem.

I would argue that the paper could benefit greatly from comparing against all related works. This should include both a theoretical comparison of the proposed approach (e.g. which parts of TC are novel), and an empirical evaluation of how this translates to training speed, generalization, etc.



### Concern 2: Safety Implications of Thought Cloning

One of the two main rationales from the introductory section is that thought cloning could make the resulting agent more safe.
In S 3.5 / Fig.5, authors evaluate a thought-based safety technique that they call "precrime intervention" and find that:
> remarkably, Precrime Intervention almost entirely eliminates all unsafe behaviors, thereby demonstrating the promising potential of TC agents in advancing AI safety

While this sounds intuitively correct, there seems to be (at least) two caveats that could result in a less safe system due to the false sense of safety.

First, under the formulation proposed in Section 2, Thought Cloning is incentivized to copy any cognitive errors from the expert, including lies, false rationalizations and general ignorance.
Using the car driver example from L48-49, an agent could (falsely) argue that it is legal to ignore some traffic rule in this specific circumstance. It is easy to imagine that human "expert" drivers will avoid "testifying against themselves" without a dedicated system to control for that. A sufficiently powerful language model can mimic their plausible explanations to mislead the passenger.

Second, if the proposed system is improved upon -- either directly with Reinforcement Learning or indirectly with manual updates, the agent would be incentivised to produce explanations that satisfy the user, so as to avoid being stopped. Note that these explanations do not have to be true - they simply have to be convincing. This can be seen as a special case of the "stop button problem" (see, e.g. , https://arxiv.org/pdf/1611.08219.pdf or https://www.lesswrong.com/posts/wxbMsGgdHEgZ65Zyi/stop-button-towards-a-causal-solution )

In other words, the initial system has a risk of inheriting potential flaws of (human) experts and any improvements to the system run a risk of incentivising less truthful explanations. From my (limited) point of view, it appears to early to claim that TC precrime intervention can truly increase the real-world system safety. One way to improve this would be to add a dedicated safety analysis, formulate and prove safety guarantees.

**Questions:**

> we can improve AI agents by training them to think like humans do.

Disclaimer: i am no expert on the subject matter, feel free to ignore the question.

In the beginning of the paper, (L5-L6) it is presented a fact that humans think in a language.
To the best of my knowledge, there is no scientific consensus that all humans think in a language, or that humans think only in language - though I am no an expert on that account. Is this really true? If so, please cite the relevant work (e.g. in S1). If not, please paraphrase the sentence to avoid accidentally misleading the reader.

**Limitations:**

Authors have adequately adressed most of the limitations of their work. As for the sociental impact, I have highlighted several safety concerns for the proposed system in the "weakness" section.

---

> ### Author Rebuttal · Authors · 2023-08-09
>
> Thank you for your review, and for noting our method is reasonable and supported by our evaluations and arguments.
>
> We sincerely believe our paper presents a significant and beneficial contribution that would enrich the ML community upon publication. Reflecting upon your concerns, we believe they either have been fixed, or are areas for future work rather than fundamental flaws that should prevent publication. Overall we would have predicted a higher score based just on your comments alone. Your current score recommends not publishing this work, but we hope you will keep an open mind to reconsidering in light of our revisions, replies, and since two other reviewers rated it as 7 (accept) and 8 (strong accept).
>
> We have attempted to address each of your concerns, and significantly improved the manuscript as a result.
>
> **1. Concern about the lack of comparison with prior work**
>
> ***“There are several previous papers that also attempt to teach an agent to think using natural langauge. Authors address planning in natural langauge in section 4.1, but there is a line of work that is not covered by planning.”***
>
> We will add the following text in the related works section to address this:
>
> “There are many works that use LLMs to improve the ability of agents to act [Wang 2023], but their approaches are quite different. They generate thoughts in addition to actions [Yao et al. 2022] or let LLMs replan based on new information [Madaan et al. 2023, Shinn et al. 2023], but none directly learn to think while acting by imitating human thought data. Thus, unlike Thought Cloning, they do not benefit from learning from human thought demonstrations how to do things like plan, replan, create high-level goals and the subgoals required to achieve them, and the many other benefits of thinking intelligently during acting.”
>
> Please note that the area and the work you mentioned “Wang et al, Describe, Explain, Plan and Select: Interactive Planning with Large Language Models Enables Open-World Multi-Task Agents” is covered in L308.
>
> ***“I would argue that the paper could benefit greatly from comparing against all related works. This should include both a theoretical comparison of the proposed approach (e.g. which parts of TC are novel), and an empirical evaluation of how this translates to training speed, generalization, etc.”***
>
> We compared TC to what we feel is the closest baseline (behavioral cloning). As just discussed, there are other works that may seem related (since they use LLMs to think in some ways), but they are actually quite different, and direct comparisons are thus apples to oranges and not very scientifically informative. However, we did add a performance comparison to one similar related work (Think Before You Act, Mezghani et al., 2023), which also learns from action + language data in the same domain (BabyAI). Thought Cloning outperforms it, showing that the main idea of Thought Cloning (imitating human thinking) is beneficial. (See Table 1 in new results PDF)
>
> **2. Concern about the Safety Implications of Thought Cloning**
>
> ***“under the formulation proposed in Section 2, Thought Cloning is incentivized to copy any cognitive errors from the expert, including lies, false rationalizations and general ignorance.”***
>
> Thank you for highlighting this potential concern. We added the following text as a result:
>
> “As occurs with LLM pretraining and other forms of Behavioral Cloning, Thought Cloning could inadvertently inherit undesirable human flaws, such as speaking falsehoods, providing false yet persuasive rationalizations, or being biased. Alignment techniques are being constantly improved to address these challenges [Ouyang et al. 2022]. However, even improving AI agent safety up to the level of a (flawed) human would be a major advance, even if the resulting system is not perfect. Additionally, a distinguishing feature of Thought Cloning is it provides the ability to interpret and prevent these flaws from culminating into actions, making TC a more favorable approach in this regard.”
>
> ***“if the proposed system is improved upon -- either directly with Reinforcement Learning or indirectly with manual updates, the agent would be incentivised to produce explanations that satisfy the user, so as to avoid being stopped. Note that these explanations do not have to be true - they simply have to be convincing.”***
>
> We agree, but that is a reason to avoid training the system to not be stopped by precrime intervention. One should instead train agents to do desirable things and avoid undesirable things directly, and reserve precrime interruption as a safety mechanism (and not training reward function). Additionally, one could use ever-smarter LLMs (or, better, multi-modal models like vision-language models) to monitor the thoughts and actions to attempt to detect such issues, a capability that is only possible because Thought Cloning provides interpretable access to the agent's thoughts. We will add these nice insights to the paper. Thanks!
>
> ***“In other words, the initial system has a risk of inheriting potential flaws of (human) experts and any improvements to the system run a risk of incentivising less truthful explanations. From my (limited) point of view, it appears to early to claim that TC precrime intervention can truly increase the real-world system safety. One way to improve this would be to add a dedicated safety analysis, formulate and prove safety guarantees.”***
>
> Our paper contains arguments for and experimental analyses on the effectiveness of precrime intervention, which can serve to inspire others to advance this method and or test it in real-world scenarios. We’ll ensure the writing does not guarantee it will be effective, but instead (like much helpful scientific publications) simply provides evidence of a new, promising method.
>
> **3. It is debatable whether humans exclusively think in language.**
>
> Addressed in the Response to All Reviewers.

---

> ### Comment · Reviewer_HBeY · 2023-08-14
> **Reviewer should reconsider their score**
>
> I feel that the score given in this review is excessively low given the content of the review. Even granting that there are thorny issues with the safety applications, I don't think the reviewer has brought up any fatal criticism that indicate "technical flaws, weak evaluation, inadequate reproducibility and incompletely addressed ethical considerations." Moreover, the authors have responded cogently to the comments. Therefore, I encourage this reviewer to raise their score, or otherwise indicate why their concerns have not been addressed.

---

### Official Review · Reviewer_gnSR · 2023-07-27

**Soundness:** 3 good
**Presentation:** 2 fair
**Contribution:** 3 good
**Rating:** 8
**Confidence:** 4

**Summary:**

This paper presents an approach incorporates discrete intermediate-level descriptions and goals  to train RL agents to perform higher level actions. These intermediate-level descriptions and goals are described in natural language, and the authors refer to them "thoughts" that the RL agent learns which then influences a desired action, in a process called Thought Cloning (TC). The paper contrasts this approach with conventional Imitation Learning that rely on Behavioral Cloning (BC). The paper claims that TC learns faster than BC and generalizes much better than BC across cognitive difficulty and behavioral difficulty.

**Strengths:**

Originality: This paper appears to propose a novel method, but I am not certain of that.

Quality: The collected experimental data and the collection of synthetic training data are are all sound, and several valid metrics are provided to support the papers claim, and this is a complete, but, I believe, flawed work due to the construction of the ablation study.

Significance: If the ablation study can be properly addressed and modified to show a clear advantage of TC over BC, then this paper would be very significant.

**Weaknesses:**

Writing: Prose used in the abstract and introduction is superfluous at times. Many sections speculate why the proposed method is superior supported by biological comparison, but the paper lacks evidence for any such comparison. The introduction could possibly be halved in length and still convey the most important aspects of this paper. Specifically, lines 47-81 could easily be condensed.

Quality: The ablation study performed, as I understand it, is the weakest component of this paper. The implementation of TC w/o Imitating Thought is described opaquely, and leaves a lot of inference to the reader. As I understand it from the paper, and copy of the TC architecture is made, however the control signals for Next Thought and Thought History are left null. While this does create a model with an equivalent number of parameters as the original TC, only the parameters for the action agent are trained.

Clarity: See above issues with interpreting the ablation study.

Experiment Design: Comparing BC to TC is an imperfect comparison as noted by the paper and hence the inclusion of the TC w/o Imitating Thought, however, this model does not adequately address difference between model performance either. My intuition tells me that TC significantly outperforms BC because it has access to more training data (the intermediate thoughts) and comprises of more parameters which then allows the model to train faster and generalize better. The paper does not do an adequate job to address this concern. A possible new experiment that could address this is to ensure training data is equivalent between the models, and allow the BC model to approximate the same number of parameters as TC. Without this additional comparison, it cannot be confidently known where the true cause for the improvement.

**Questions:**

What is the architecture of the TC w/o imitating Thought and what are its control signals?

**Limitations:**

A major limitation to this approach mentioned in the introduction is the availability of applicable thought data.

---

> ### Author Rebuttal · Authors · 2023-08-09
>
> Thank you for your detailed review and acknowledging the strengths of our work, including its novelty and saying it is ***“very significant”*** (provided your requested experiments confirm TC outperforms BC, which they all did!).
>
> We have addressed each of your concerns, significantly improving the manuscript as a result. We feel this paper makes an important contribution the ML community will benefit from if published. Your current score is low and likely will prevent publication. We hope you will keep an open mind to changing your score in light of our improvements, and since two other reviewers rated the paper a 7 (accept) and 8 (strong accept).
>
> ***“The implementation of TC w/o Imitating Thought is described opaquely ... As I understand it …[a] copy of the TC architecture is made, however the control signals for Next Thought and Thought History are left null. While this does create a model with an equivalent number of parameters as the original TC, only the parameters for the action agent are trained.”***
>
> All parameters were trained in the 'TC w/o Imitating Thought' control (described in L158-162).
>
> We apologize it was not clearer. We will add the following to make it more clear.
>
> “we introduce an ablation variant called TC w/o Imitating Thought to demonstrate that TC’s superiority is not solely due to its larger number of parameters. This variant’s architecture is identical to the TC architecture, except for a minor architectural difference where the latent vector from the upper level is directly input to the lower level as thought. This adjustment is necessary because this variant is not trained on the Thought Cloning loss, so we do not have per-word supervision. To train these parameters, we thus need to train them based on how these “thoughts” contribute to the final performance (i.e. they are trained end-to-end). If we sampled words (as in Thought Cloning), we could not train these parameters end-to-end because hard sampling of words is non-differentiable, so gradients could not flow backward through this operation. Thus, we make one small change in order to allow the parameters to be trained via SGD, which is to pass the logits of the Upper-level Component directly into the Lower-level Component, which is a differentiable operation. We feel this is the closest and fairest control possible to Thought Cloning, allowing virtually the same architecture and the same number of parameters, but not including the Thought Cloning innovation of exploiting human thought data.”
>
> ***"My intuition tells me that TC significantly outperforms BC because it has access to more training data (the intermediate thoughts) and comprises of more parameters which then allows the model to train faster and generalize better."***
>
> We designed the TC w/o Imitating Thought control to address this concern, allowing the baseline to have the same number of parameters (and architecture) as Thought Cloning. We hope now that you know these parameters were trained, your main objection to the paper is removed. However, for completeness, we try to address it in another way. Instead of holding the architecture the same, we create a BC control with a more canonical BC architecture, but with the same number of parameters as TC. Results show it does not perform nearly as well as TC. (see new results PDF)
>
> To address if “has access to more training data (the intermediate thoughts)”, we will add this text:
>
> “One might argue that TC gets more data (one set of actions and one set of words per episode), and thus that a proper BC control is to give BC 2x data. A counter is that such a control is unnecessary because noticing and harnessing this additional (often free) and heretofore ignored data stream is a central contribution of this paper, so showing that using this data improves things is the main comparison to be made. However, for completeness, we ran an experiment giving BC twice as much data: results show BC-2xData is still far slower to learn and has significantly worse performance at convergence.” (See new results PDF)
>
> Since you said TC’s performance advantage might come from having extra parameters or more data, we wanted to make sure it did not come from having both. We tried BC with 2x data and a pure BC architecture with the same number of parameters as TC, and it too underperformed TC.
>
> Also, a prior work “Think Before You Act [Mezghani et al. 2023]” had a similar number of episodes of actions and language data in the same domain, and their model has more parameters than ours. TC also outperforms them. (See new results PDF)
>
> You wrote “If the ablation study can be properly addressed and modified to show a clear advantage of TC over BC, then this paper would be very significant.”
>
> The original control, a new comparison to prior work, and all of the new experiments you requested do indeed show a clear advantage of TC over BC (dramatically faster learning and better performance at convergence), providing strong evidence that the advantages of TC are not solely due to additional training data or model capacity. Given that you said that would make this paper “very significant” were this true, we hope you are open to substantially increasing your score. Thank you for considering it!
>
> ***"Many sections speculate why the proposed method is superior supported by biological comparison, but the paper lacks evidence for any such comparison."***
>
> Thank you for pointing this out. We will be happy to revise the writing. While we used human thinking as a source of inspiration, and we do think it suggests this direction is powerful, one could jettison that analogy and our method would stand alone on its empirical results (higher performance, debugability, safety benefits, etc.). We will eliminate superfluous language and make these issues clearer in the paper.
>
> ***“A major limitation to this approach mentioned in the introduction is the availability of applicable thought data.”***
>
> Addressed in the Response to All Reviewers.

---

> > ### Comment · Reviewer_gnSR · 2023-08-21
> >
> > Thank you for addressing my concerns completely. I believe the paper should be accepted, and have changed my score to reflect such.
> >
> > The various control architectures matching number of parameters and amount of data leaves me with confidence that the TC architecture is the cause of the improvement.
> >
> > One additional request (that I feel would further strengthen the paper) could the authors address why "TC w/o Imitating Thought" continues to achieve a much lower success rate in comparison to the various control architectures? My initial reaction is to say that architecture overall is deficient for a true comparison.
> >
> > I understand that this is the "closest and fairest control possible," (and how I might have constructed such a control) but it's puzzling why "TC w/o Imitating Thought" never achieved similar success rates, when every other control model did.

---

### Author Rebuttal · Authors · 2023-08-09

We are deeply grateful to the reviewers for their comprehensive evaluations and thoughtful feedback on our work.

We are encouraged by the reviewers' positive comments including:
- “The approach is innovative, and could be **highly impactful** once it is scaled up.” (HBeY) “The idea of generating actions conditioned on thoughts (descriptive texts for actions) is simple but effective.” (9iZD)
- “If the ablation study can be properly addressed and modified to show a clear advantage of TC over BC, then **this paper would be very significant**.” (gnSR) [The new, requested ablation studies do show such a clear advantage]
- “The paper makes **a strong empirical case** that their approach works” (HBeY) “The paper contains a reasonable evaluation of the proposed idea… Typical caveats are addressed with a reasonable ablation analysis.” (QGuE) “The collected experimental data and the collection of synthetic training data are all sound.” (gnSR)
- “The application to AI safety and interpretability is interesting and increases the significance of the contribution.” (HBeY) “TC also enhance the interpretability, since human can see descriptive texts for each actions.” (9iZD)
- “TC is trained faster than BC… TC may improve the ability to generalize to out-of-distribution environments” (9iZD)
- “The paper is well-written and easy to follow.” (QGuE) “This paper is very well written and organized” (9iZD)

We feel this paper makes an important, helpful contribution that the ML community will benefit from if published. **We appreciate that two reviewers rated the paper a 7 (accept) and 8 (strong accept).** But the two other scores are low and likely will prevent publication. We hope all reviewers will keep an open mind to increasing your scores in light of our improvements to improve the chances that the ML community will be able to benefit from learning about this work

In response to your requests, we have produced many new results (see results PDF), all of which strongly empirically support the benefits of Thought Cloning. We also improved the paper in the ways you suggested. Here is a quick summary of the major revisions:
- We clarify that all parameters were in fact trained in the “Thought Cloning w/o Imitating Thought” control, and thus there was already a behavioral cloning control for the number of parameters (and architecture), eliminating one reviewer’s major objection. While this was described in the original paper, we have modified the paper to make the description of this control much clearer. For completeness, we also performed additional experiments requested by reviewer gnSR, all of which show that the superiority of TC does not come from a larger number of parameters or from (arguably) having more training data. Instead, it is the main idea of Thought Cloning that drives the performance gains.
- We modified the language in the paper’s motivation making it clear that it is simply a hypothesis (and one up for debate) that humans think in natural language (and have added citations that the claim is debatable). We also point out that this biological comparison is simply motivation, and that the method can independently stand on its empirical results even if one does not believe in this hypothesis or analogy.
- We addressed the safety concern raised by reviewer QGuE.
- We expanded and improve comparisons to prior work.
- We've expanded the explanation for Eq. 1 and explained the error bars more clearly in the corresponding figure captions.

The revisions have made the paper stronger and we deeply thank all the reviewers for their help. Below we respond to common questions, and the reviewer-specific responses answer unique comments.

**1. Regarding our motivation text saying humans think in natural language:**

Reviewer QGuE, HBeY: **It is debatable whether humans exclusively think in language.**

Thanks for mentioning this. While (as you noted) this issue is not essential for judging whether Thought Cloning is a valuable method (since it performs well), the claim is a central inspiration for the work. To recognize the claim is debated, we will update the text as follows:

“While it remains debated whether humans exclusively think in language [Grandchamp et al. 2019, Alderson-Day et al. 2015], some argue that natural language is intricately woven into our thought processes [Premack 2004, Chomsky 2006], conferring distinct cognitive advantages. [Pinker 2003]”

**2. Concern about the availability of thought data:**

Reviewer 9iZD: ***“How can we effectively collect thoughts for actions to train TC agents?”***

Reviewer GnSR: ***“A major limitation to this approach mentioned in the introduction is the availability of applicable thought data.”***

We apologize for the confusion. It appears there may have been a misinterpretation regarding our stance on the availability of thought data. Our introduction intended to convey the opposite: that thought data is widely available, but it's an avenue that hasn't been sufficiently explored by the community. As an example, we could collect action demonstrations from unlabeled Youtube videos with VPT [Baker et al. 2022, Fan et al. 2022] and pair them with related video transcripts (the closed captions of the video where people narrate what they are doing and why while acting). This gives us a dataset encompassing both human cognition and action [Fan et al. 2022]. We proposed this approach in L62-66 and L91-96 in our paper. A contribution of this paper is introducing a method that can mine that heretofore under-exploited treasure trove of data. There are also other (albeit more expensive) ways to get data, such as human volunteers, contractors, employees, or vision-language models that create such narrations for videos.

---

### Decision · Program_Chairs · 2023-09-21

**Decision:**

Accept (spotlight)

**Comment:**

Imitate both the behaviors of an agent and any language that agent generates while performing those actions. While other work attempts to add LLMs to RL agents in various ways, this method focuses on learning to generate the same "thoughts" as a task is carried out.

Reviewers were unanimous that this is an interesting direction that could lead to better imitation learning algorithms and richer settings. Raising the abstraction level of reasoning performed by agents is likely to pay off.

If I could give a conditional acceptance to this manuscript, I would. Throughout the manuscript the authors make the point that they imitate human thoughts. But they do not. They imitate language generated from rule-based descriptions of a planner. There is a big gap between the two, which is the underlying reasons for reviewers pushing back on the general applicability of this work. No dataset where humans describe their thoughts in this manner exists. The authors contend that one could use narrated YouTube videos, but there is no evidence that this would work. It is also why this work leads to complex discussions about the nature of thoughts.

It is far simpler to describe what the manuscript actually does: the method does not mirror human thoughts. It mirrors self-reports from a planner in natural language that describe what the planner was doing. I would encourage the authors to update their manuscript to make this clear. This actually makes their method stronger: any time you can have a planner generate descriptions you can apply their method to more quickly align your agent with that planner; perhaps in a way that generalizes better. The applicability of the method as originally described, while sounding far more grandiose, is actually very limited.

That being said, the method is novel, it is a promising method to incorporate language into RL, and so as the reviewers point out, it deserves to be published.